# Molecular Identification of Enteric Viruses in Domestic Animals in Northeastern Gabon, Central Africa

**DOI:** 10.3390/ani13152512

**Published:** 2023-08-03

**Authors:** Linda Bohou Kombila, Nadine N’dilimabaka, Déborah Garcia, Océane Rieu, Jéordy Dimitri Engone Ondo, Telstar Ndong Mebaley, Larson Boundenga, Matthieu Fritz, Léadisaelle Hosanna Lenguiya, Gael Darren Maganga, Eric M. Leroy, Pierre Becquart, Illich Manfred Mombo

**Affiliations:** 1Unité Émergence des Maladies Virales (UEMV), Département de Virologie, Centre Interdisciplinaire de Recherches Médicales de Franceville (CIRMF), Franceville BP 769, Gabon; bohoukombilalinda@gmail.com (L.B.K.); nadinendilimabaka@yahoo.fr (N.N.); telstarlunique@gmail.com (T.N.M.); gael_maganga@yahoo.fr (G.D.M.); 2Département de Biologie, Université des Sciences et Techniques de Masuku (USTM), Franceville BP 941, Gabon; 3Institut de Recherche pour le Développement (IRD), Maladies Infectieuses et Vecteurs, Écologie, Génétique, Évolution et Contrôle (MIVEGEC) (Université de Montpellier—IRD 224–CNRS 5290), 34394 Montpellier, France; deborah.garcia@ird.fr (D.G.); oceane.rieu@ird.fr (O.R.); fritz.matthieu@iage-france.com (M.F.); eric.leroy@ird.fr (E.M.L.); pierre.becquart@ird.fr (P.B.); 4Unité des Infections Rétrovirales et Pathologies Associées (UIRPA), Centre Interdisciplinaire de Recherches Médicales de Franceville (CIRMF), Franceville BP 769, Gabon; engonejordy@gmail.com; 5Unité de Recherche en Écologie de la Santé (URES), Centre Interdisciplinaire de Recherches Médicales de Franceville (CIRMF), Franceville BP 769, Gabon; boundenga@gmail.com; 6Faculté des Sciences et Techniques, Université Marien Ngouabi, Brazzaville BP 69, Congo; hosannelenguiya@gmail.com; 7Institut National Supérieur d’Agronomie et de Biotechnologies (INSAB), Université des Sciences et Techniques de Masuku (USTM), Franceville BP 913, Gabon

**Keywords:** astroviruses, enteroviruses, caliciviruses, vesivirus, small ruminants, dogs, zoonoses, rural areas

## Abstract

**Simple Summary:**

Enteric viruses cause gastroenteritis in humans and animals, but have also been associated with several extraintestinal diseases. These viruses affect a wide range of vertebrate species, including birds and mammals. Frequent and close contact between humans and animals can potentially lead to the emergence of zoonoses. Approximately 60% of zoonotic diseases in humans are spillovers from wildlife. Preventing future outbreaks of emerging zoonotic diseases calls for better description of the viruses that circulate in domestic animals, because they are located at the human/wildlife interface. We therefore screened for the presence of astroviruses, enteroviruses, and caliciviruses—three of the main viral families causing enteric diseases both in humans and animals—in goats, sheep and dogs living in villages in northeastern Gabon. We identified the presence of astroviruses in goats, a calicivirus in a dog, and enteroviruses in all three species. All detected viruses were animal-related, but not those from wildlife. However, we showed that some human-pathogenic enteroviruses infect goats and dogs. Therefore, further studies are required to better understand the role of domestic animals as amplifiers of reverse zoonotic viruses.

**Abstract:**

Astroviruses (AstVs)*,* enteroviruses (EVs), and caliciviruses (CaVs) infect several vertebrate taxa. Transmitted through the fecal–oral route, these enteric viruses are highly resistant and can survive in the environment, thereby increasing their zoonotic potential. Here, we screened for AstVs, EVs, and CaVs to investigate the role of domestic animals in the emergence of zoonoses, because they are situated at the human/wildlife interface, particularly in rural forested areas in Central Africa. Rectal swabs were obtained from 123 goats, 41 sheep, and 76 dogs in 10 villages located in northeastern Gabon. Extracted RNA reverse-transcribed into cDNA was used to detect AstVs, EVs, and CaVs by amplification of the RNA-dependent RNA polymerase (RdRp), or capsid protein (VP1) gene using PCR. A total of 23 samples tested positive, including 17 goats for AstVs, 2 goats, 2 sheep, 1 dog for EVs, and 1 dog for CaVs. Phylogenetic analyses revealed that AstV RdRp sequences clustered with sheep-, goat-, or bovine-related AstVs. In addition, one goat and two sheep VP1 sequences clustered with caprine/ovine-related Evs within the *Enterovirus G* species, and the CaV was a canine vesivirus. However, human-pathogenic Evs, EV-B80 and EV-C99, were detected in goats and dogs, raising questions on the maintenance of viruses able to infect humans.

## 1. Introduction

Zoonoses are infectious diseases that are naturally transmitted between humans and animals [1]. Approximately 60% of infectious diseases in humans have a zoonotic origin and 70% of them are emerging diseases [1,2,3]. Zoonotic diseases occur at the human/animal interface where humans and domestic or wild animals are in close interaction [4,5]. Many zoonotic infections are neglected because they mainly affect populations that live in close proximity with animals in rural areas where sanitary systems are poor or scarce [6]. Over the last decades, zoonotic infections such as Ebolavirus disease, Zika, and the SARS-CoV-2 pandemic have demonstrated the threat zoonoses can represent to global public health [7,8], sometimes also causing high economic losses. The surveillance of zoonotic diseases needs to be reinforced and focused on the detection of pathogens circulating in wildlife and domestic animals. For instance, the SARS-CoV-2 virus that recently emerged acquired the ability to infect a large range of animals fearing a reverse transmission of the virus to human population [9].

Astroviruses (AstVs), enteroviruses (Evs), and caliciviruses (CaVs) are small positive-strand RNA viruses that infect a large spectrum of wild and domestic animals, such as apes, minks, felines, cattle, birds, dogs, and pigs, and humans. They are transmitted directly via the fecal–oral route or indirectly via contaminated food, surfaces, and water, and they replicate primarily in the gastrointestinal tract [10,11,12]. Infections of AstVs, Evs, and CaVs are commonly associated with diarrhea in humans and animals. Nonetheless, they can cause mild to severe extraintestinal infections, such as respiratory diseases, neurological disorders including myelitis or acute flaccid paralysis, encephalitis, nephritis, and skin diseases. However, there are cases of asymptomatic animals carrying these viruses [13], thus domestic and wild animals do not always get sick form the presence of these viruses.

Astroviruses (AstVs), first discovered in 1975, are small non-enveloped viruses belonging to the genera *Mamastrovirus* and *Avastrovirus* within the family *Astroviridae* [14]. *Mamastrovirus* comprises AstVs detected in mammals, and *Avastrovirus* includes those identified in birds. AstVs were initially thought to be highly species-specific [15], but evidence of some AstVs detected in various host species indicates the ability of these viruses to cross the species barrier. Moreover, the recent identification of two novel groups of human AstVs—namely Melbourne (AstV-MLB) and Virginia/human-mink-ovine (AstV-VA/HMO), highly divergent from classic human AstVs, and genetically closer to animal AstVs—has suggested their putative zoonotic origin [16,17,18].

Enteroviruses are members of the genus *Enterovirus* within the *Picornaviridae* family that consists of 15 species such as *Enterovirus A* to *L* and *Rhinovirus A* to *C* [19]. Evs are known to infect humans and belong to species *Enterovirus A* to *D*, which also include Evs detected in various primate species [20,21,22,23]. In addition, immunological studies have demonstrated the presence of neutralizing antibodies against bovine Evs in humans [24]. Thus, these findings suggest that interspecies transmissions between human and primates are frequent, and bovine Evs may have zoonotic potential.

Caliciviruses (CaVs) belong to the *Caliciviridae* family, which consists of 11 recognized genera including *Bavovirus*, *Norovirus*, *Sapovirus*, *Lagovirus*, and *Vesivirus* (https://ictv.global/taxonomy, accessed on 9 April 2023) [25]. Only noroviruses and sapoviruses can infect humans. Despite the identification of CaVs in a wide range of animal species, none of the noroviruses or sapoviruses are known to be zoonotic [26]. Unlike other CaVs, vesiviruses are the only members of the family that can readily cross the host species barrier [27].

AstVs, Evs, and CaVs are highly stable in the environment [12,28], and can thus be infectious for a long period of time. Many studies screening for enteric viruses in animals have reported interspecies transmission between humans and animals. To understand the emergence of infectious diseases, the reservoirs, and/or sensitive hosts, it is crucial to investigate the carriage of viruses in domestic animals, providing useful information to better prevent the emergence of zoonotic diseases. Domestic animals and humans live in close proximity in the same environment. Especially in rural areas, small herds of small domestic ruminants are common, and dogs are pets or raised for hunting. Domestic species may therefore act as intermediaries at the human/wildlife interface and thus contribute to the spillover of wildlife-borne zoonotic viruses to humans, leading to the emergence of diseases. Here, we screened for AstVs, Evs, and CaVs in small ruminants and dogs living in villages in northeastern Gabon, Central Africa, to determine if domestic animals can act as intermediate hosts in zoonotic transmissions of AstVs, Evs, and CaVs. This study reports the first evidence of AstVs, EVs, and CaVs in goats, sheep, or dogs in Gabon.

## 2. Materials and Methods

### 2.1. Study Sites and Sample Collection

This present study is part of a larger multicentric project named ‘EBO-SURSY’ led by the WHOA on “Capacity building and surveillance for viral hemorrhagic fevers”, and more broadly on zoonotic viruses. We focused this research on Gabon, in the department of Zadié, an administrative subdivision of Ogooué-Ivindo province, situated in northeastern Gabon, where Ebola virus epidemics were reported between 1994 and 2002. Thus, this study was carried out in 10 villages found in a tropical rainforest area, mainly composed of primary forests, along two roads leading out of Mekambo, the capital of Zadié: the Mekambo-Mazingo and Mekambo-Ekata roads (Figure 1). Samples were taken from small ruminants (goats and sheep) and dogs from November 2018 to August 2022.

After obtaining consent from the owners, animals were handled by veterinarians and rectal swabs were collected. The swabs were stored in liquid nitrogen at the health center in Mekambo, before final storage at −80 °C at CIRMF in Franceville, in southeastern Gabon.

### 2.2. RNA Extraction and Reverse Transcription

Rectal swabs were suspended in 600 µL of phosphate saline buffer (PBS) and shaken at room temperature at 7000 rpm for 30 min. RNA was extracted from the suspension using the Nucleospin RNA Virus, Mini kit for viral RNA from cell-free fluids (Macherey-Nagel, Düren, Germany), following the manufacturer’s recommendations. Extracted RNA was then reverse-transcribed into cDNA using Superscript IV Reverse Transcriptase (Invitrogen, Illkirch, France) in a final volume of 20 µL. Specifically, 10 µL of RNA in a mix consisting of 1 µL dNTPs, 1 µL random hexamer and 1 µL of DNase-free water was incubated at 65 °C for 5 min followed by 1 min on ice. A second mixture, made up of 4 µL of 5X Superscript IV Buffer, 1 µL of DTT (100 mM), 0.5 µL of RNAse Out, 0.5 µL of enzyme and 1 µL of DNAse-free water, was added to the previous mix. The reverse-transcription program was set to 23 °C for 10 min, 50–55 °C for 10 min, and 80 °C for 10 min.

### 2.3. Enteric Virus Detection

Enteric virus screening was performed using PCR or nested PCR with Platinum Taq DNA Polymerase (Invitrogen) along with primer sets targeting AstVs, EVs, and CaCVs (Table 1). For nested PCR, both rounds were performed in a final reaction volume of 25 µL. For the first round, the mix consisted of 5 µL of cDNA, 2.5 µL of 10× reaction buffer, 0.75 µL of MgCl_2_ (50 mM), 0.5 µL of dNTPs, 1 µL of bovine serum albumin (BSA) (1 µg/µL), 1 µL of each primer (10 µM), and 0.1 µL of Platinum Taq Enzyme. The second round was performed using 5 µL of the PCR product from the first round and the mixture included the components and volumes from the first round with the exception of BSA (not included). Amplification programs for AstVs, EVs, and CaVs were set up as previously described [21,29,30], without the reverse-transcription step for the first round. PCR products were visualized on a 1.5% agarose gel after electrophoresis, then amplicons were sent to Eurofins Genomics (Ebersberg, Germany) for Sanger sequencing in both directions.

### 2.4. Phylogenetic Analyses

Sequences were assembled and edited using the Seqman program implemented in LaserGen 7 (DNAstar). Then, AstV, EV, and CaV sequences obtained in this study were compared with a dataset of sequences retrieved from GenBank using the Basic Local Alignment Search Tool (BLAST) (NCBI, Bethesda, MD, USA). Multiple sequence alignments were carried out using the ClustalW algorithm implemented in MEGA 11 software (version 11.0.13) [34]. The trees were constructed using the maximum likelihood method in PhyML available online (http://phylogeny.lirmm.fr/phylo_cgi/index.cgi, accessed on 23 March 2023) with the GTR model of branch support and 100 bootstrap replicates [35].

The sequences obtained in this study are available in GenBank under accession numbers OR188789-OR188805 for AstVs, OR188806-OR188810 for EVs and OR188811 for CaV.

## 3. Results

### 3.1. Detection of Astroviruses, Enteroviruses and Caliciviruses

For the 240 rectal swabs used for the detection of enteric viruses in this study, 123 were from goats, 41 from sheep, and 76 from dogs collected in the 10 villages located along two main roads in Zadié. The number of goats, sheep, and dogs sampled in each village is summarized in Table 2. A total of 23 domestic animals tested positive for enteric viruses with an overall prevalence of 9.6%. Specifically, 17 (7.1%) of all goats tested positive for AstVs; 5 (2.1%) tested positive for EVs: 1 dog, 2 goats, and 2 sheep; and 1 dog (0.4%) tested positive for CaVs. No animal tested positive for two or more enteric viruses.

### 3.2. Genetic Diversity of Astroviruses, Enteroviruses and Caliciviruses

Phylogenetic analyses were performed to genetically characterize the sequences of AstVs, EVs, and CaVs and determine if sequences obtained in this study were related to any known virus capable of infecting humans.

For AstVs, none of the RNA-dependent RNA polymerase (RdRp) sequences obtained clustered with known human AstVs of the classic group or groups of potentially zoonotic origin, such as AstV-HMO-A to -C and MLB-1 to -3. Indeed, all the RdRp sequences from the AstV-positive samples were split among three distinct lineages: lineages a, b, and c (Figure 2). Lineage a consisted of two AstV sequences detected in goats from the same village. Lineage b consisted of five sequences of AstVs detected in goats from villages located along the Mekambo-Mazingo road. Finally, lineage c consisted of AstV sequences detected in goats from villages along both roads: the Mekambo-Mazingo road and the Mekambo-Ekata road. The nucleotide identities between lineages ranged from 58.0 to 60.6% between lineages a and b, 62.9 to 63.2% between lineages a and c, and 65.8 to 66.8% between lineage b and c. Goat-related AstV sequences of lineage a belonged to the *Mamastrovirus 13* clade. In this clade, which contains AstVs associated with neurotropic disease in ruminants, the sequences were genetically close to the ovine OvAstV strain UK/2013/ewe/lib01454 (accession number LT706531). UK/2013/ewe/lib01454 was identified in a Welsh Mountain ewe in the United Kingdom [36]. The sequences shared 93% nucleotide identity with this OvAstV strain. For lineage b, our sequences grouped with sequences related to diverse host species of family Bovidae, including water buffalo, cattle, and goat. The sequences were similar to bovine AstV BSRI (KP264970), which was detected in cattle in the USA [37]. The sequences showed nucleotide identities ranging from 77.5 to 80.1%. By contrast, sequences of lineage c, comprised solely of AstV-infected goats in Gabon, were highly divergent from other AstVs found in ruminants, with nucleotide identities ranging from 60.3 to 74.9%. Capsid sequences are needed to determine if the lineage c is a new genotype of genus *Mamastrovirus*.

For EVs, the capsid protein (VP1) sequences were obtained from all five samples that tested positive (two from sheep, two from goats, and one from dog). The phylogenetic analyses showed that two sequences from sheep (NTO-53-OV and NTO-54-OV) and one from goat (MDB-190-CP) were genetically similar (95.0 to 98.0% nucleotide identity). The sequences clustered with the *Enterovirus G* species, which consists of EVs identified in swine, goats, and sheep. Specifically, within EV-G, three sequences were phylogenetically close to ovine EV 2019-00927 isolate NA (OV176440) responsible for myelitis in lambs in Austria [38]. The sequences shared 75 to 76% nucleotide identity. In addition to the identification of animal-related EVs, the sequences of the second goat EV (MDB-32-CP) and the dog EV (ZOU-111-CN) were related to human EVs. MDB-32-CP grouped with *Enterovirus B* and ZOU-111-CN grouped with *Enterovirus C*. Specifically, the MDB-32-CP sequence was assigned to an EV-B80 strain with which it shared from 84.9 to 89.5% nucleotide identity with other EV-B strains (Figure 3). ZOU-111-CN clustered with EV-C99 strains and shared nucleotide identities ranging from 79.5 to 89.5%.

Finally, the phylogenetic analyses based on the RdRp gene showed that the CaV sequence detected in a dog (IMB-009-CN) did not belong to *Norovirus* or *Sapovirus*, which are the only genera within the *Caliciviridae* known to infect humans. Instead, the sequence clustered with the *Vesivirus* genus (Figure 4). Within *Vesivirus*, the sequence clustered with other dog vesiviruses and was similar to a canine vesivirus (MF327134) detected in a US military dog sampled in 1968; however, it was reported only in 2018 [39]. The sequences shared 92.0% nucleotide identity.

## 4. Discussion

At the human/animal interface, especially in rural areas, domestic animals may constitute intermediaries for the spillover of zoonotic wildlife-borne viruses. This could lead to the emergence of zoonotic diseases. The screening of enteric viruses in domestic animals of villages in northeastern Gabon allowed for the detection of astroviruses, enteroviruses, and calicivirus (specifically vesivirus). The detection rates varied according to the viruses and the animal species.

### 4.1. Astroviruses

AstVs were only detected in goats. After other studies in bats (4.6%), rodents (2.4%), and primates (1.5%), the detection of AstVs in goats offered further information on their circulation and their animal hosts in Gabon [29,40,41]. The phylogenetic analyses based on RdRp revealed that the sequences of goat AstVs highlighted in this study were genetically diverse within the *Mamastrovirus* genus, being distributed among three different lineages (a, b, and c). Although some human AstVs from the AstV-HMO and AstV-VA groups have been suggested to be zoonotic and related to animal AstVs (i.e., related to mink, bats and sheep), none of these lineages are genetically close to human AstVs.

Nevertheless, lineages a and b clustered with AstVs that were detected in animals with neurological disorder or encephalitis as clinical manifestations. Moreover, AstVs from lineage a clustered with AstVs belonging to *Mamastrovirus 13*, a clade of AstVs known to be associated with neurological disorders in small ruminants and cattle in the United Kingdom, Germany, Switzerland, Norway, and China [36,42]. In addition to AstVs able to cause neurological disorders, AstVs of lineage b, which clusters bovine, caprine, and water buffalo AstVs, were genetically close to a bovine AstV detected in cattle with bovine respiratory disease (BRD). Contrary to the neurotropic AstVs in animals, the association of AstVs with BRD has not been clearly determined. Indeed, other viruses such as bovine adenovirus 3, bovine influenza virus D, or bovine rhinitis viruses, also responsible for BRD, were detected in the same herd of cattle and some of them were coinfected with several viruses. Therefore, these coinfections may have exacerbated the symptoms [37]. However, veterinarians recorded no diarrhea or neurological manifestations in any goats during the swab collection phase.

Finally, comparing AstVs from lineages a and b detected in animals from the same village or from along the same road, AstVs from lineage c were detected in goats from villages along both roads. Small ruminants are often given as gifts during various events including traditional weddings, births, or religious ceremonies. Thus, it is possible that animals were transferred between villages, thus leading to the spread of AstVs. In addition, AstVs of lineage c formed a monophyletic group of AstV sequences in Gabonese goats, divergent from other ruminant AstVs. Further investigations are thus needed to better characterize this monophyletic group. However, the classification of AstVs being based on the capsid gene [15], a phylogeny based on the complete capsid gene is needed to confirm this hypothesis.

### 4.2. Enteroviruses

EVs were detected in goats (2/123, 1.6%), sheep (2/41, 4.9%), and dogs (1/76, 1.3%). This is the first report of detection of EVs in these animals in Gabon. The phylogenetic analyses based on the VP1 sequences showed that the goat- and sheep-related EV sequences belonged to the *Enterovirus G* species. This species, together with the *Enterovirus F* species, includes caprine/ovine EVs [43,44]. Caprine/ovine EVs are emerging viruses, but little is known about their epidemiological aspects, distribution, epidemic patterns, and other affecting factors.

The phylogenetic analyses revealed that the goat and sheep VP1 EV sequences were genetically close (95.0 to 98.0% nucleotide identity) suggesting that the circulation of the same EVs among these domestic animals. Sheep and goats roam freely over the same area in villages, which can foster the crossing of the species barrier of the same EV between these small ruminant host species.

Two human EVs were also detected: EV-B80 in a goat (MDB-32-CP), belonging to *Enterovirus B*, and EV-C99 in a dog (ZOU-111-CN), belonging to *Enterovirus C.* To our knowledge, this is the first report of a human EV in goat. EV-B80 was identified for the first time in 2007 [45]. Since then, the virus has been detected in various countries worldwide including China, Kenya, India, Philippines, Cambodia, and Bangladesh [46,47,48,49,50]. To date, there is little data regarding the epidemiology and pathologies caused by this virus. By contrast, although this is the first identification of EV-C99 in dogs, EV infections in dogs have been extensively investigated from the 1960s to the 1990s. These investigations sought to understand how polioviruses and enteroviruses are perpetuated for a better global eradication of polioviruses [51,52]. In these studies, detection of neutralizing antibodies in blood and viral isolation from feces or nasal swabs showed that human EVs, including coxsackieviruses, polioviruses and human echoviruses, can infect dogs. [51,52]. The first identification of an EV-C99 strain dates back to 2000. Subsequently, other strains have been identified in healthy children and those with diseases worldwide [53,54]. Regarding zoonoses, various strains of EV-C99 have been described in non-human primates in the Republic of Congo and Bangladesh [55,56]. Therefore, transmission may have occurred through the consumption by animals of contaminated water or food by human feces. The detection of human EVs in goats and dogs may pose risks for human health. Even though no outbreak of EV-B80 and EV-C99 has been recorded, infections may be responsible for acute flaccid paralysis in children [57,58]. Further investigations are needed to describe the epidemiology and the distribution of EV-B80 and EV-C99 in children and animals in Gabon.

### 4.3. Caliciviruses

A CaV was detected in one dog (IMB-009-CN). The sequence obtained clustered with canine CaVs of the genus *Vesivirus*. Vesiviruses were identified for the first time in 1932 in domestic pigs with vesicular disease in the USA. Since then, vesivirus infections have been reported in various animal species, including sea lions, minks, cats and dogs [27]. The genus *Vesivirus* consists of only two recognized species: *Vesicular exanthema swine virus* and *Feline calicivirus*. Other vesiviruses including canine vesivirus are still unclassified as species (https://ictv.global/taxonomy, accessed on 9 April 2023). The first canine vesivirus was isolated in the early of the 1980s [59]. Within the canine vesiviruses, our sequence IMB-009-CN was genetically close to the sequence of the canine vesivirus 3–68 (MF328134) strain that infected a US military dog. This finding confirmed that vesiviruses are common in dogs worldwide. Although there is no evidence of a canine vesivirus infection in humans, some studies have reported the zoonotic potential of vesiviruses. Infections with the San Miguel sea lion have been reported to cause vesicular exanthema on the hands after a laboratory incident, and on the face of a field biologist working with marine mammals. In addition, vesiviruses infections have been suspected to cause illness in human after the observation of antibodies anti-*Vesivirus* together with *Vesivirus* viremia in human with hepatitis with unknown etiology [27,60]. Nonetheless, no other subsequent studies have reported vesivirus infections in humans [61].

## 5. Conclusions

In conclusion, this study reported a diversity of enteric viruses with the detection of AstVs, EVs, and CaVs in domestic animals and provided information regarding their circulation in Gabon. The detection of animal-related AstVs, EVs, and CaVs showed a low zoonotic risk for humans. In contrast, even though none of the viruses detected were related to wildlife, two human EVs (EV-B80 and EV-C99) were detected in goats and dogs, respectively. These findings demonstrate that domestic animals can harbor human enteric viruses, thereby constituting a risk of zoonotic transmission. Further studies are necessary to understand whether goats and dogs can serve as intermediate hosts, becoming sources of EV-B80 and EV-C99 spillback to humans, and whether these viruses circulate naturally in these animals.

## Figures and Tables

**Figure 1 animals-13-02512-f001:**
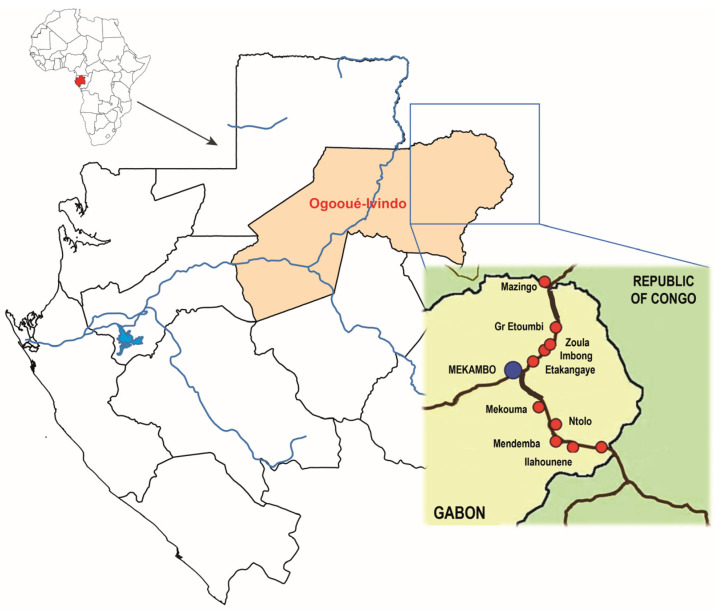
Map of the Gabonese villages (red circles) where samples were collected from goats, sheep, and dogs.

**Figure 2 animals-13-02512-f002:**
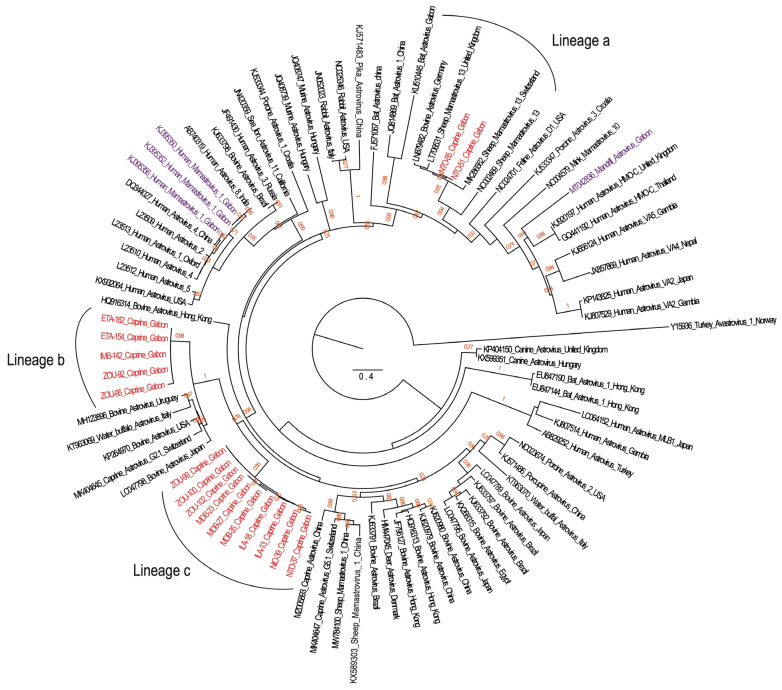
Phylogenetic tree of astroviruses (AstVs) based on approximately 422 bp of a fragment of the RNA-dependent RNA polymerase (RdRp) gene. Sequences sampled from goats are indicated in red, those in purple indicate other AstVs detected in Gabon. The maximum-likelihood tree was constructed with 100 replications; bootstrap values greater than 0.6 are shown at the nodes. Accession numbers, species and countries are shown.

**Figure 3 animals-13-02512-f003:**
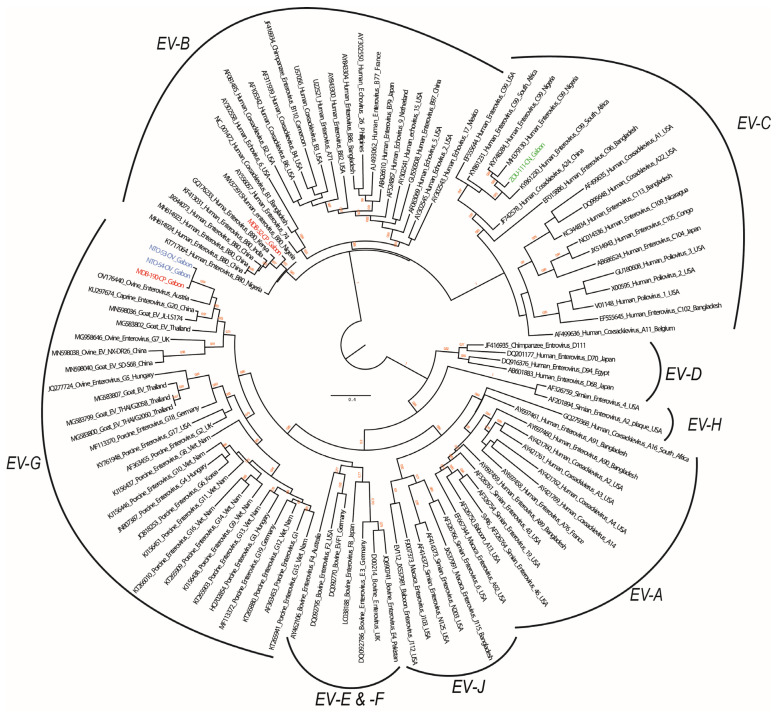
Phylogenetic tree of enteroviruses (EVs) based on approximately 371 bp of the capsid protein (VP1) gene. Sequences sampled from goats in this study are indicated in red, blue indicates sequences obtained from sheep and green, the sequence obtained in a dog. Bootstrap values lower than 0.6 are not shown.

**Figure 4 animals-13-02512-f004:**
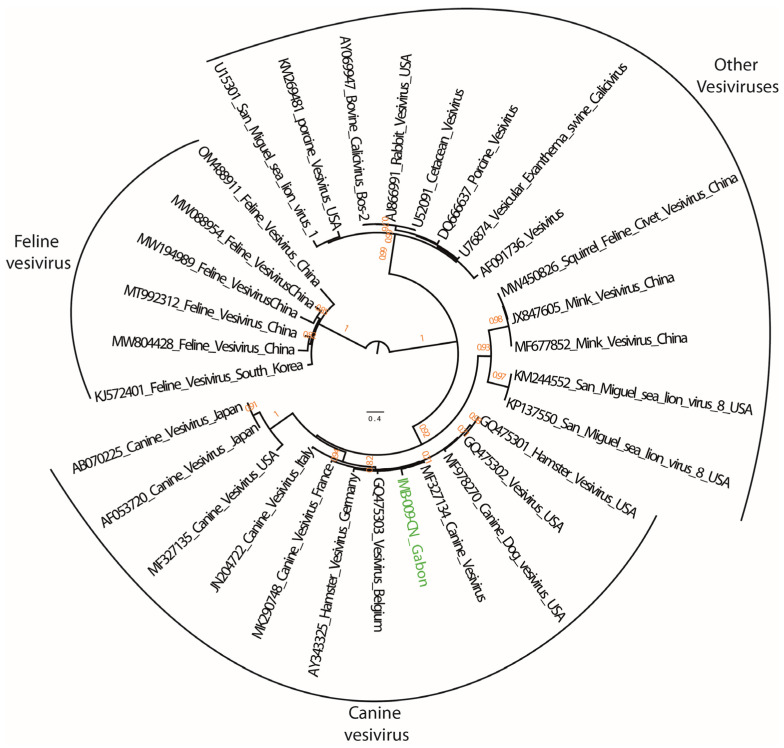
Phylogenetic tree of vesiviruses based on the 320 bp of the RNA-dependent RNA polymerase (RdRp) gene. Color code is the same as for Figure 3. Only bootstraps ≥ 0.6 are indicated above the branches.

**Table 1 animals-13-02512-t001:** Sequences of primers used for the detection of astroviruses, enteroviruses, and caliciviruses.

Target Virus Name	Gene	Primer Name	Primer Sequence (5′-3′)	Size	References
Astrovirus	RdRp	FWD1	GARTTYGATTGGRCKCGKTAYGA	422 bp	[31]
		FWD2	GARTTYGATTGGRCKAGGTAYGA		
		RVS1	GGYTTKACCCACATNCCRAA		
		FWD3	CGKTAYGATGGKACKATHCC		
		FWD4	AGGTAYGATGGKACKATHCC		
Calicivirus	RdRP	P289	TGACAATGTAATCATCACCATA	319–331 bp	[30]
		P290	GATTACTCCAAGTGGGACTCCAC		
Enterovirus	VP1	222	CICCIGGIGGIAYRWACAT	371 bp	[32,33]
		224	GCIATGYTIGGIACICAYRT		
		AN88	TACTGGACCACCTGGBGGNAYRWACAT		
		AN89	CCAGCACTGACAGCAGYNGARAYNGG		

**Table 2 animals-13-02512-t002:** Results from the screening for astroviruses, enteroviruses, and caliciviruses in small ruminants and dogs.

Road	Village	Astroviruses Positive/Tested			Enteroviruses Positive/Tested			Caliciviruses Positive/Tested		
		Goats	Sheep	Dogs	Goats	Sheep	Dogs	Goats	Sheep	Dogs
Mekambo - Mazingo	Etakangaye	2/18	-	0/1	0/18	-	0/1	0/18	-	0/1
	Imbong	1/16	0/5	0/13	0/16	0/5	0/13	0/16	0/5	1/13
	Ibéa	-	-	0/2	-	-	0/2	-	-	0/2
	Zoula	5/9	0/6	0/11	0/9	0/6	1/11	0/9	0/6	0/11
	Grand-Etoumbi	0/8	0/13	0/18	0/8	0/13	0/18	0/8	0/13	0/18
Mekambo - Ekata	Mékouma	-	-	0/7	-	-	0/7	-	-	0/7
	Ntolo	4/18	0/2	0/21	0/18	2/2	0/21	0/18	0/2	0/21
	Mendemba	3/26	-	-	2/26	-	-	0/26	-	-
	Ilahounènè	2/8	0/8	0/2	0/8	0/8	0/2	0/8	0/8	0/2
	Ekata	0/20	0/7	0/1	0/20	0/7	0/1	0/20	0/7	0/1
Total		17/123	0/41	0/76	2/123	2/41	1/76	0/123	0/41	1/76

## Data Availability

The data are available from the corresponding author upon reasonable request. The sequences obtained in this study are available in GenBank under accession numbers OR188789-OR188805 for AstVs, OR188806-OR188810 for EVs and OR188811 for CaV.

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
