# Peer review of "Molecular Identification of Enteric Viruses in Domestic Animals in Northeastern Gabon, Central Africa"

_animals, 2023, doi:10.3390/ani13152512_

Round 1

Reviewer 1 Report

Kombila et al. in their paper titled “Identification of Enteric Viruses in Domestic Goats, Sheep and 2 Dogs Living in Villages Located in Northeastern Gabon, Central Africa” screened for AstVs, EVs and CaVs to investigate the role of domestic animals in the emergence of zoonoses. Using RT-PCR and sequencing analyses, they found a total of 23 samples tested positive including 17 goats for AstVs, 2 goats, 2 sheep 47 and 1 dog for EVs and 1 dog for CaVs. Phylogenetic analyses also revealed the phylogenetic status for these identified AstV, EVs, and CaVs. These results provide the new epidemiological data for the above three enteric virus infections.

1.      Line 62  “……in rural? Areas” should be changed to “…. in rural areas”.

2.      Line 66 “needs to be reinforced and focus on…” should be changed to “needs to be reinforced and focused on….”.

3.      Phylogenetic analysis for the identified enterovirus is only based on VP1 of 371 bp sequence, it might not reflect the true picture of the phylogenetic status for these identified viruses.

no comment

Reviewer 2 Report

The study has great relevance for Public Health Surveillance.

I presented some suggestions in order to improve the qualification of the authors' initiative.

I made suggestions for changes in the title of the article and small changes in the text of the introduction that I thought were most appropriate.

Other minor corrections into material and methods and results, including figure captions in the article.

In the discussion several suggestions for adequacy, as well as in the conclusion.

References were well chosen and cited throughout the text.

Congratulations on the job!

Reviewer 3 Report

Anthropozoonoses constitute a major problem for domestic animals and human inhabiting areas in close proximity with animal farms. The manuscript is therefore scientifically sound and of high public health importance. Methodologically, the study is correct and well explained, whereas the manuscript is well written. Nevertheless, there are some issues that have to be addressed / modified before considered for publication. These issues are generally related with lack of discussion of the origin of detected viruses as well as lack of presenting the health status of the examined animals.

Specifically, a distinct part (probably 4.4) should be added in the discussion regarding the origin of viruses based on the phylogenetic results. It is indeed occasionally difficult to propose such assumptions, however at least some hypotheses can be provided.

Line 62: Delete question mark “?” after “rural”

Lines 75-78: The authors should add, however, that there are cases of asymptomatic animals carrying these viruses (https://doi.org/10.1186/s12985-022-01787-1), thus domestic and wild animals do not always get sick form the presence of these viruses

In the introduction, the authors should also add some info regarding the study area, i.e. how many animals are reared, how often are zoonoses observed, etc. I would recommend a paragraph to be added. 

Please also add in the Materials and Methods a Table indicating the health status of the sampled animals. Animals may categorised in three categories, healthy, ill and mild. This is important 

Round 2

Reviewer 3 Report

Despite the limitation of not knowing the health status of the animals, the authors replied sufficiently to all my comments and therefore I recomend publication